# Local Muscle Endurance and Strength Had Strong Relationship with CrossFit^®^ Open 2020 in Amateur Athletes

**DOI:** 10.3390/sports9070098

**Published:** 2021-07-06

**Authors:** Ramires Alsamir Tibana, Ivo Vieira de Sousa Neto, Nuno Manuel Frade de Sousa, Caroline Romeiro, Adriana Hanai, Hiury Brandão, Fábio Hech Dominski, Fabricio Azevedo Voltarelli

**Affiliations:** 1Graduate Program in Health Sciences, Faculty of Medicine, Federal University of Mato Grosso (UFTM), Cuiabá 78060-900, Brazil; ntr.carolineromeiro@gmail.com (C.R.); voltarellifa@gmail.com (F.A.V.); 2Laboratory of Molecular Analysis, Graduate Program of Sciences and Technology of Health, University of Brasilia, Brasilia 72220-900, Brazil; ivoneto04@hotmail.com; 3Laboratory of Exercise Physiology, Faculty Estacio of Vitoria, Vitoria 29092-280, Brazil; nunosfrade@gmail.com; 4Graduate Program on Physical Education, Catholic University of Brasilia, Brasilia 71966-700, Brazil; adrianac.hanai@gmail.com (A.H.); hiuryedf@gmail.com (H.B.); 5Laboratory of Sport and Exercise Psychology, Human Movement Sciences Graduate Program, College of Health and Sport Science of the Santa Catarina State University (UDESC), Florianópolis 88080-400, Brazil; fabiohdominski@hotmail.com

**Keywords:** functional fitness training, athletic performance, exercise testing

## Abstract

This study analyzed the relationship between anthropometric measures, cardiorespiratory capacity, strength, power, and local muscle endurance with performance in the CrossFit^®^ Open 2020. For this, 17 volunteers (6 women) (29.0 ± 7.2 years) completed, on separate weeks, tests for body composition (dual-energy X-ray absorptiometry), maximal oxygen consumption (2 km row test), muscle strength (one repetition maximum (1 RM) back and front squat, isometric peak torque), muscle power (1 RM snatch and clean and jerk) and muscle endurance (Tibana test), which were compared with performance during the CrossFit^®^ Open 2020. Specific tests of localized muscular endurance and muscle strength had the strongest relationship with performance in the CrossFit^®^ Open 2020. On the other hand, the percentage of fat and cardiorespiratory capacity were not significantly correlated with CrossFit^®^ Open 2020 workout performance. Coaches and practitioners should therefore utilize these findings to assess physical fitness and organize the distribution of the training session based on less developed physical needs, in order to ensure an appropriate physiological adaptation for a given competition.

## 1. Introduction

CrossFit^®^ is considered as a constantly varied, high-intensity, functional movement training aiming at increasing work capacity across multiple physical domains (endurance, strength, flexibility) using several functional movements [1]. Therefore, different types of workout sessions, which are known as Workout of the Day (WOD), combine different exercises and tasks, such as cycling, running, rowing, Olympic weightlifting, power weightlifting and gymnastic-type exercises [1].

Competitive CrossFit^®^ often consists of two stages, the online qualifier (multiple unfamiliar workout challenges that are completed over the course of some weeks) from which the competitors with the best online results qualify for regional events (South America, North America, Africa, Asia etc.). The CrossFit^®^ Open is an online competition officially managed by CrossFit Inc, and is one of the largest sports events in the world, with more than 239,106 participants in the 2020 event.

The specific training for the CrossFit^®^ Open tends to be different for regional events (South America, North America, Africa, Asia etc.), due to the characteristics of that competition (low overload is usually used in resistance exercises and tasks with longer duration). In addition, only rowing ergometer is traditionally prescribed as a cyclic exercise (usually with low volume).

Despite the increase in popularity, there is a paucity of literature on the topic of CrossFit performance. Previous research has investigated the relationship between non-specific tests such as anthropometric profile [2], cardiorespiratory capacity [3], muscle strength [3], and muscle power [3], with performance in practitioners and athletes of CrossFit. For example, Martínez-Gómez et al. [3] evaluated the relationship between muscle strength (full squat) and performance at the CrossFit^®^ Open 2017. The authors showed that strength and power indices measured in a squat test were positively associated with CrossFit^®^ performance. However, according to Butcher et al. [4], performance in CrossFit is different from most sports where it is possible to predict and evaluate the performance of an athlete with tests of muscular strength, physiological variables, and aerobic and anaerobic powers (test on treadmill and Wingate, respectively). In CrossFit, although the tests are partially associated with performance in some tests (CrossFit open and benchmarks), these tests have no specificity with the variety of movements and repetitions during the workouts (calisthenics, strength, and endurance).

The workouts in the previous year (2019) had no characteristics of muscle strength or cardiovascular capacity. On the other hand, the athletes performed several repetitions of specific exercises, such as a gymnastic component (e.g., burpees, strict handstand push-ups or walks, chest to bar pull-ups, toes-to-bar, and bar muscle-ups) with relatively light resistance exercises (e.g., wall ball, thruster, clean, snatch, dumbbell overhead lunge, dumbbell box step-ups). However, it is unknown whether a specific test of local muscle endurance is associated with performance in the CrossFit^®^ Open. Prior physical performance screening might provide efficient data for analysis and important feedback to athletes and coaches to determine key performance predictors in a particular competition. This information could be valuable for evidence-based strategies during competitions and to identify possible deficient performance.

Therefore, the current study aimed to analyze the relationship between anthropometric measures, cardiorespiratory capacity, and the variables of strength, power, and a specific test of muscular endurance with performance in the CrossFit^®^ Open 2020. We hypothesized that significant correlations would be found between the variables analyzed and performance in the CrossFit^®^ Open 2020.

## 2. Materials and Methods

### 2.1. Participants

In total, 17 volunteers (6 women) (29.0 ± 7.2 years) were recruited. All subjects were free of injury or known illnesses, were not using performance enhancing drugs, and had a minimum of 6 months of CrossFit experience. Participants were advised to sleep six to eight hours the night before the tests, maintain regular nutritional and hydration habits, avoid intense exercise 48 h prior to the sessions, and avoid smoking, alcohol, and caffeine consumption 24 h before a session. All subjects provided informed consent and the study was approved by the University Research Ethics Committee for Human Use (2.698.225/Universidade Estácio de Sá/UNESA/RJ) and conformed to the principles of the Helsinki Declaration on the use of human participants for research.

### 2.2. Experimental Design

The present study followed a cross-sectional design. All participants performed the baseline assessments two weeks prior to the CrossFit^®^ Open 2020 (five workouts for 5 weeks) (October–November 2019). Figure 1 shows schematic illustration of the methodological steps in the present study.

### 2.3. Baseline Assessments

The participants performed, on separate weeks, body composition (dual-energy X-ray absorptiometry), maximal oxygen consumption (2 km row test), muscle strength (1 RM back squat, 1 RM front squat and isometric peak torque, Biodex System 3), and muscle power measures (1 RM snatch and 1 RM clean and jerk), as well as a specific test of muscle endurance (Tibana test). Baseline assessments were performed two weeks before the CrossFit^®^ Open 2020.

### 2.4. Anthropometric Measurements

The anthropometric measurements were made in the morning after an overnight fast, with the subjects wearing light clothing and no shoes. The participants were weighed on a Filizola^®^ digital scale (Curitiba, PR, Brazil; capacity of 180 kg) to the nearest 0.1 kg. Each subject was measured in accordance with the standard methods proposed by the International Society for the Advancement of Kinanthropometry [5].

### 2.5. Body Composition

The body composition evaluation, including body fat percentage and fat-free mass, was performed through dual energy X-ray absorptiometry (General Electric-GE model 8548 BX1L, year 2005, Lunar DPX type, software Encore 2005; Rommelsdorf, Germany) with a coefficient of variation of 0.9–1.1%. Briefly, the examinations included a complete body scan of the participants in a supine position. The equipment was continuously regulated and operated by a technically trained health professional. Legs were secured by nonelastic straps at the knees and ankles, and the arms were aligned along the trunk with the palms facing the thighs.

### 2.6. Maximal Oxygen Consumption

The 2 km row test was performed as previously described by Jensen et al. [6]. All exercise tests were performed on the same rowing ergometer (model E; Concept 2, Morrisville, VT, USA). The subjects individually adapted their preferred stroke rate and drag factor in tests and during the warm-up protocol. The standardized warm-up for the 2 km time trail was 4 min of easy rowing; 4 × 1 min with 10, 15, 20, and 10 hard strokes for each minute; and 2 min of easy rowing. After a short rest while the oxygen uptake equipment was mounted, the 2 km all-out time trail was performed.

Before and during the test period, athletes were monitored by ECG (Digital Electrocardiogram, Micromed Brazil, São Paulo, SP, Brazil) to detect possible anomalies that would make it impossible for the volunteers to continue the test or study. Heart rate (HR) was measured with a Polar H10 (Polar Electro Oy, Kemple, Finland). Gases were analyzed using the ERGOPc ELITE/Metasofit 3 device (Brasília, DF, Brazil). The VO_2peak_ and HR values adopted were the highest achieved during the test. All procedures were followed by a cardiologist.

### 2.7. Muscle Strength and Power Measures

Participants performed one repetition maximum (1 RM) test for back squat, front squat, snatch, and clean and jerk according to procedures recommended by the National Strength and Conditioning Association [7,8] with 48 h rest intervals between sessions to minimize the muscle fatigue effects and pain. All randomized tests were performed with a barbell (20/15 kg) and weights (1–25 kg) (Pood Fitness^®^, Brasília, DF, Brazil). The test protocol consisted of a brief general warm-up on a bike or indoor rowing ergometer, followed by eight repetitions at 50% of estimated 1 RM (according to the prior loads used by members in their training exercise routines). Following a 1 min rest, they performed three repetitions at 70% of estimated 1 RM. Then, the participants completed three to five attempts with 3 to 5 min rest intervals between each attempt, with progressively heavier weights until the 1 RM was established. All testing sessions took place between 2 p.m. and 3 p.m. after lunch and under a controlled standardized temperature (25 °C). During this exercise period, standard instructions regarding the procedures of the test protocols and the appropriate execution of the exercise technique were supplied by a qualified and experienced investigator.

### 2.8. Isometric Strength

The subjects were seated in a knee extension chair that allowed for isometric force assessment. Subjects were restrained in the chair with straps secured around their chest, abdomen, and hips. Their arms remained crossed during testing. Three maximal isometric contractions of knee extension were performed on the isokinetic dynamometer (Biodex System 3, Biodex Medical Systems, Shirley, NY, USA). The knee joint was fixed at 60° for performance of the isometric contraction for a period of 5 s, with the subject seated and hips fixed at a 90° angle [9]. A 3 min recovery period was provided between isometric contractions. The highest value was considered as the maximum isometric torque [9].

### 2.9. Local Muscle Endurance

The local muscle endurance test applied was the Tibana test, which included the conclusion of four distinct rounds of work, each separated by 2 min of rest (Figure 2). Specifically, the rounds involved 4 min of as many rounds as possible (AMRAP) of five thrusters (60 kg for men and 43 kg for women) and 10 box jumps over (round 1); 4 min of AMRAP of 10 power clean (60 kg for men and 43 kg for women) and 20 pull-ups (round 2); 4 min of AMRAP of 15 shoulder to overhead (60 kg for men and 43 kg for women) and 30 toes to bar (round 3); and 4 min of AMRAP of 20 calories of rowing and 40 wall ball (9 kg for men and 6 kg for women) (round 4) [10,11].

### 2.10. CrossFit^®^ Open 2020

The specific details of the five workouts of the day (WODs) used in this study, known as 20.1, 20.2, 20.3, 20.4, and 20.5, are briefly explained below:20.1: Participants had 15 min to complete 10 rounds of the following exercises: 8 ground-to-overheads (43 kg men; 29.5 kg women) and 10 bar-facing burpees.20.2: Participants had 20 min to complete as many rounds as possible of 4 dumbbell thrusters (22.5 kg for men and 16 kg for women—dumbbells), 6 toes-to-bars, and 24 double-unders.20.3: Participants had 9 min to complete 21 deadlifts, 102/70 kg, 21 handstand push-ups 15 deadlifts, 102/70 kg, 15 handstand push-ups, 9 deadlifts, 102/70 kg, 9 handstand push-ups, 21 deadlifts, 143/93 kg, 15 m handstand walk, 15 deadlifts, 143/93 kg, 15 m handstand walk, 9 deadlifts, 143/93 kg and 15 m handstand walk.20.4: Participants had 20 min to complete 30 box jumps, 60/50 cm, 15 clean and jerks, 43/29.5 kg, 30 box jumps, 60/50 cm, 15 clean and jerks, 61/39 kg, 30 box jumps, 60/50 cm, 10 clean and jerks, 84/52 kg, 30 single-leg squats, 10 clean and jerks, 102/66 kg, 30 single-leg squats, 5 clean and jerks, 125/80 kg, 30 single leg squats, and 5 clean and jerks, 143/93 kg.20.5: Participants had 20 min to complete 40 muscle-ups, 80-cal row, 120 wall-ball shots, (9 kg ball to 3 m/6 kg ball to 2.75 m).

### 2.11. Statistical Analysis

The results are expressed as mean value ± standard deviation (SD). The Shapiro–Wilk test was applied to check for parametric distribution of study dependent variables. Simple Pearson’s r correlations were utilized to determine the associations between the results of CrossFit^®^ Open 2020 benchmarks and anthropometric, strength, cardiorespiratory, and performance measures. The magnitude of the correlations was classified as: *r* ≤ 0.1 trivial; 0.1 < *r* ≤ 0.3 small; 0.3 < *r* ≤ 0.5 moderate; 0.5 < *r* ≤ 0.7 large; 0.7 < *r* ≤ 0.9 very large; *r* > 0.9 almost perfect [12]. The level of significance was *p* ≤ 0.05 and SPSS version 20.0 (Somers, NY, USA) software was utilized.

## 3. Results

### 3.1. Anthropometric, Strength, Cardiorespiratory, and Performance Data Presentation

The anthropometric and cardiorespiratory measurements are shown in Table 1 and the strength measurements are shown in Table 2. The participants achieved 252.3 ± 31.7 repetitions (men) and 257.7 ± 36.0 repetitions (women) during the Tibana test.

### 3.2. CrossFit^®^ Open 2020 Data and Correlations

There was a statistically significant correlation between age and CrossFit^®^ Open 2020.1 (*r* = 0.95; *p* < 0.01), CrossFit^®^ Open 2020.2 (*r* = −0.98; *p* < 0.01), and CrossFit^®^ Open 2020.4 (*r* = −0.85; *p* = 0.03) for women. No correlations were observed between age and CrossFit^®^ Open 2020 benchmarks for men. Specific tests of localized muscular endurance and muscle strength had the strongest relationship with performance in the CrossFit^®^ Open 2020. There were no significant associations between anthropometric measures and CrossFit^®^ Open 2020 for men or women.

The correlations between CrossFit^®^ Open 2020 benchmarks and the strength, cardiorespiratory, and performance measures for men are shown in Table 3. CrossFit^®^ Open 2020.1 and CrossFit^®^ Open 2020.5 were only correlated with the Tibana test score (very large negative correlation). There was a statistically significant correlation between Tibana test scores and C&J values with CrossFit^®^ Open 2020.2 (very large positive correlation) and between Tibana test scores, FS, C&J, and TS values with CrossFit^®^ Open 2020.3 (very large positive correlation). Lastly, only strength measures presented a statistically significant correlation with CrossFit^®^ Open 2020.4 (very large and almost perfect positive correlation).

The correlations between CrossFit^®^ Open 2020 benchmarks and the strength, cardiorespiratory, and performance measures for women are shown in Table 4. CrossFit^®^ Open 2020.1 was correlated with the Tibana test score, snatch, and C&J values (very large negative correlation). There was a statistically significant correlation between Tibana test scores, VO_2max_, snatch, and C&J values with CrossFit^®^ Open 2020.2 (very large positive correlation), between FS, C&J, and TS values with CrossFit^®^ Open 2020.3 (very large and almost perfect positive correlation), and between Tibana test scores, FS, snatch, C&J and TS values with CrossFit^®^ Open 2020.4 (very large and almost perfect positive correlation). Lastly, only FS and TS presented a statistically significant correlation with CrossFit^®^ Open 2020.5 (very large and almost perfect negative correlation).

## 4. Discussion

This study aimed to analyze the relationship between anthropometric measures, cardiorespiratory capacity, and the variables of strength, power, and a specific test of muscular endurance with performance in the CrossFit^®^ Open 2020. Confirming the initial hypothesis, specific tests of localized muscular endurance and muscle strength had the strongest relationship with performance in the CrossFit^®^ Open 2020. On the other hand, contrary to previous studies, the percentage of fat [2] and cardiorespiratory capacity [3] were not significantly correlated with CrossFit^®^ Open 2020 workout performance.

Local muscle endurance had a strong relationship with athletic performance in the CrossFit^®^ open 2020, which might be related to the test specificities used in the current study. The Tibana test consists of a variety of exercises and movements which place greater stress on gymnastic components (pull-ups and toes-to-bar) and resistance exercises (thruster, clean, shoulder to overhead, and wall ball), which are requisites for almost all athletic endeavors and fitness components involved in CrossFit. Current data reinforce the use of the Tibana test or another specific test as a key tool for physical performance screening of practitioners of functional fitness training.

In a prior review, Suchomel et al. [13] explained potential benefits of muscular strength in athletic performance. It is well established that greater muscular strength is associated with enhanced force-time characteristics, including rate of force development, external mechanical power, and magnitude of potentiation, besides general skill performance, such as jumping, sprinting, and change of direction [13]. These adaptations may clarify why relative total muscle strength can be vastly influential in improving an individual’s overall performance during CrossFit^®^ workouts. Furthermore, in some CrossFit Open workouts there are tests with strength characteristics which can be advantageous for stronger athletes.

An important characteristic in relation to CrossFit Open WODs is that they can vary in terms of intensity, duration, and skills, as well as physiological demands. For example, in workouts where total muscle strength was more related to performance (20.3 and 20.4), exercises (deadlift and clean and jerk) included high overload characteristics. On the other hand, in the other workouts, the most common characteristics were the large number of repetitions with a gymnastic component and lighter resistance exercises. Thus, it appears that the correlations in relation to localized muscle endurance, cardiovascular fitness, anthropometric profile, and muscle strength and power with CrossFit^®^ Open WODs are highly dependent on their characteristics and that the performance of single tests as performed in previous studies may not reflect the athletic profile of the volunteers. An effective alternative for athletes and coaches may be the holistic assessment model of athletes [14].

Accumulating evidence suggests that lower body fat can also have a beneficial effect on relevant fitness components, including endurance, strength, power, speed, balance, and agility [15,16]. The reason behind this is that additional weight creates more resistance during exercise and consequently the individual needs to work harder to maintain a certain level of performance. On the other hand, the individual may have a low body fat percentage, but show low strength levels [17]. An interesting finding of the current study is that amateur athletes displayed an acceptable body fat percentage (men 8.8 ± 5.0; women 17.1 ± 3.3%). Possibly, athletic performance is more affected by excessive body fat percentage, as demonstrated in previous studies [18,19,20].

One of the essential aspects of the practical application of the present research is to highlight that organizers of face-to-face and online competitions must know how to organize the workouts so that they really contemplate the different physical demands. For example, if a competition has a large number of events with high overloads it will favor the strongest athletes. Therefore, the competitions may not elicit the athletes with the best physical conditioning, but only the strongest athletes. Our findings will help coaches, exercise physiologists, and practitioners to assess their current training status and direct training to future competitions. For example, if the local muscle endurance is much lower than the values of competitive athletes of the modality, training can be directed to improving this physical capacity and maintaining the other capacities (strength, power, and cardiovascular).

Despite the interesting findings of this study, some limitations need to be mentioned. First, the findings are limited to a relatively small (*n* = 17) sample of convenience, our specific amateur athlete characteristics, and time frame. Second, the specific test (Tibana) does not include all gymnastics (e.g., ring and bar muscle-ups, rope climbs, pistols etc.), weightlifting (e.g., snatch), and powerlifting exercises (e.g., deadlift) that are usually performed at the CrossFit^®^ Open. Third, it should be noted that these results should be considered only for amateur athletes from a specific region and competition. Therefore, our findings may not be directly transferable to elite athletes or other experiences, which impacts the generalization and extrapolation of the obtained findings. Finally, the cross-sectional nature of this study prevents the ability to identify any causal relationship between study variables and outcomes.

## 5. Conclusions

In conclusion, we report for the first time that a specific muscle endurance test and strength had strong relationship with CrossFit^®^ Open workout performance. In contrast to our hypothesis, we did not observe any correlation between anthropometric profile and cardiorespiratory capacity with CrossFit^®^ Open performance.

From a practical perspective, coaches and practitioners can use these findings to assess physical fitness and organize the distribution of the training session based on less developed physical needs, in order to ensure an appropriate physiological adaptation for a given competition. In future studies, the focus should be on researching larger samples, different training levels, and the holistic assessment model of athletes.

## Figures and Tables

**Figure 1 sports-09-00098-f001:**
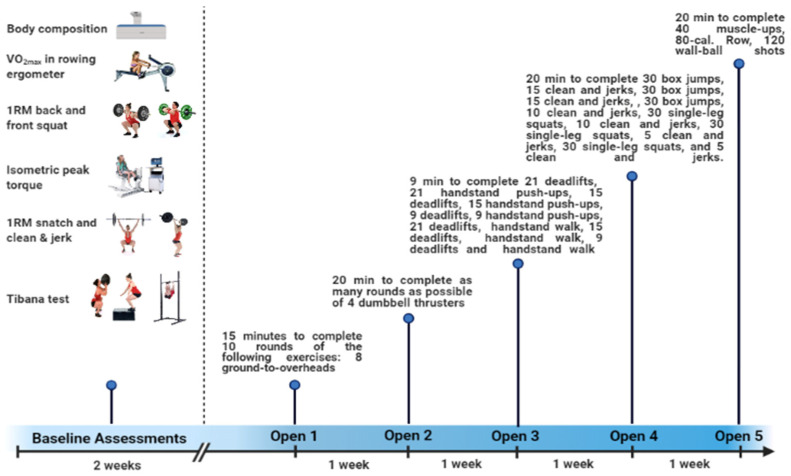
Description of study timeline.

**Figure 2 sports-09-00098-f002:**
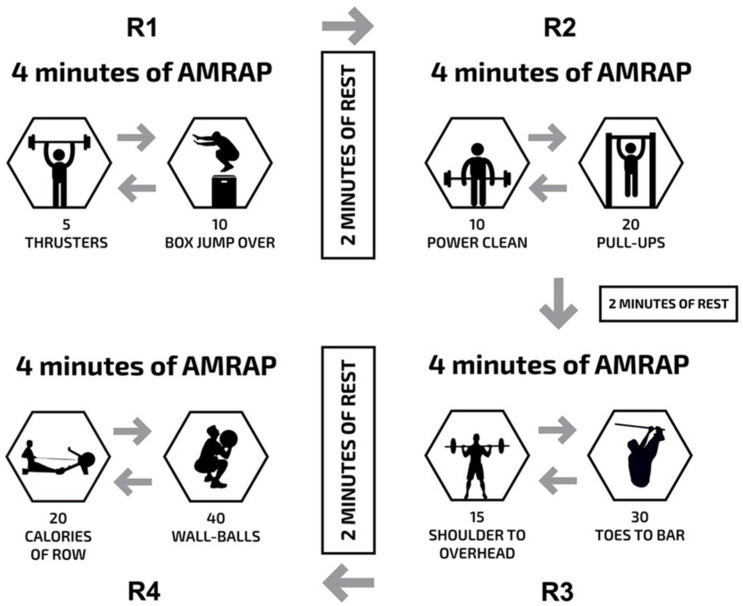
Description of the local muscle endurance test (Tibana test): 4 min of as many rounds as possible (AMRAP) of five thrusters, and 10 box jump over (round 1); 2 min of rest; 4 min of AMRAP of 10 power clean, and 20 pull-ups (round 2); 2 min of rest; 4 min of AMRAP of 15 shoulder to overhead, and 30 toes to bar (round 3); 2 min of rest; and 4 min of AMRAP of 20 calories of row, and 40 wall ball (round 4).

**Table 1 sports-09-00098-t001:** Mean ± SD of anthropometric and cardiorespiratory measurements of the participants.

Variables	Men (*n* = 11)	Women (*n* = 6)
Age, years	26.6 ± 5.7	33.3 ± 8.1
Body weight, kg	77.0 ± 3.6	58.6 ± 4.2
Body fat, %	8.8 ± 5.0	17.1 ± 3.3
Lean mass, kg	69.5 ± 4.1	48.5 ± 2.9
VO_2max_, mL·(kg·min)^−1^	49.7 ± 4.6	45.9 ± 2.1

VO_2max_, maximal oxygen consumption.

**Table 2 sports-09-00098-t002:** Mean ± SD of strength measurements of the participants.

Variables	Men (*n* = 11)	Women (*n* = 6)
Back squat, kg	145.5 ± 23.2	100.0 ± 13.7
Back squat, relative to body weight	1.9 ± 0.3	1.7 ± 0.3
Front squat, kg	129.5 ± 17.0	85.0 ± 14.5
Front squat, relative to body weight	1.7 ± 0.3	1.5 ± 0.3
Snatch, kg	91.3 ± 13.0	50.5 ± 14.0
Snatch, relative to body weight	1.2 ± 0.2	0.9 ± 0.2
Clean and jerk, kg	110.6 ± 17.0	67.7 ± 13.9
Clean and jerk, relative to body weight	1.4 ± 0.3	1.2 ± 0.2
Total strength, kg	476.9 ± 67.1	303.2 ± 49.5
Total strength, relative to body weight	6.2 ± 1.0	5.2 ± 0.9
Isometric torque, N·m	303.6 ± 25.5	202.6 ± 30.6
Isometric torque, relative to body weight	4.0 ± 0.4	3.5 ± 0.6

**Table 3 sports-09-00098-t003:** Correlations between CrossFit^®^ Open 2020 benchmarks and strength, cardiorespiratory, and performance measures in men (*n* = 11).

Variables	2020.1	2020.2	2020.3	2020.4	2020.5
VO_2max_, mL·(kg·min)^−1^	*r* = −0.54; *p* = 0.09	*r* = 0.16; *p* = 0.66	*r* = −0.19; *p* = 0.62	*r* = 0.06; *p* = 0.85	*r* = −0.30; *p* = 0.39
BS, kg	*r* = −0.28; *p* = 0.40	*r* = 0.36; *p* = 0.31	*r* = 0.59; *p* = 0.09	*r* = 0.79; *p* < 0.01 *	*r* = −0.37; *p* = 0.26
BS, rbw	*r* = −0.31; *p* = 0.36	*r* = 0.39; *p* = 0.27	*r* = 0.62; *p* = 0.08	*r* = 0.81; *p* < 0.01 *	*r* = −0.40; *p* = 0.25
FS, kg	*r* = −0.31; *p* = 0.35	*r* = 0.50; *p* = 0.14	*r* = 0.72; *p* = 0.03 *	*r* = 0.82; *p* < 0.01 *	*r* = −0.46; *p* = 0.19
FS, rbw	*r* = −0.33; *p* = 0.32	*r* = 0.51; *p* = 0.14	*r* = 0.72; *p* = 0.03 *	*r* = 0.85; *p* < 0.01 *	*r* = −0.48; *p* = 0.16
Snatch, kg	*r* = −0.43; *p* = 0.19	*r* = 0.47; *p* = 0.17	*r* = 0.60; *p* = 0.09	*r* = 0.78; *p* = 0.01 *	*r* = −0.55; *p* = 0.10
Snatch, rbw	*r* = −0.43; *p* = 0.19	*r* = 0.48; *p* = 0.16	*r* = 0.62; *p* = 0.08	*r* = 0.81; *p* < 0.01 *	*r* = −0.56; *p* = 0.09
C&J, kg	*r* = −0.50; *p* = 0.12	*r* = 0.77; *p* = 0.01 *	*r* = 0.87; *p* < 0.01 *	*r* = 0.90; *p* < 0.01 *	*r* = −0.61; *p* = 0.06
C&J, rbw	*r* = −0.49; *p* = 0.13	*r* = 0.75; *p* = 0.01 *	*r* = 0.86; *p* < 0.01 *	*r* = 0.92; *p* < 0.01 *	*r* = −0.61; *p* = 0.06
TS, kg	*r* = −0.39; *p* = 0.24	*r* = 0.54; *p* = 0.11	*r* = 0.72; *p* = 0.03 *	*r* = 0.86; *p* < 0.01 *	*r* = −0.50; *p* = 0.14
TS, rbw	*r* = −0.39; *p* = 0.23	*r* = 0.54; *p* = 0.11	*r* = 0.72; *p* = 0.02 *	*r* = 0.88; *p* < 0.01 *	*r* = −0.51; *p* = 0.13
IT, N.m	*r* = 0.20; *p* = 0.56	*r* = −0.56; *p* = 0.09	*r* = −0.58; *p* = 0.10	*r* = −0.25; *p* = 0.46	*r* = −0.03; *p* = 0.94
IT, rbw	*r* = 0.06; *p* = 0.86	*r* = −0.34; *p* = 0.33	*r* = −0.26; *p* = 0.50	*r* = 0.04; *p* = 0.92	*r* = −0.23; *p* = 0.52
Tibana test, rep	*r* = −0.73; *p* = 0.01 *	*r* = 0.83; *p* < 0.01 *	*r* = 0.74; *p* = 0.02 *	*r* = 0.51; *p* = 0.11	*r* = −0.89; *p* < 0.01 *

VO_2max_, maximal oxygen consumption; BS, back squat; FS, front squat; C&J, clean and jerk; TS, total strength; IT, isometric torque; rbw, relative to body weight; * *p* < 0.05.

**Table 4 sports-09-00098-t004:** Correlations between CrossFit^®^ Open 2020 benchmarks and strength, cardiorespiratory, and performance measures in women (*n* = 6).

Variables	2020.1	2020.2	2020.3	2020.4	2020.5
VO_2max_, mL·(kg·min)^−1^	*r* = −0.88; *p* = 0.05	*r* = −0.88; *p* = 0.05 *	*r* = −0.62; *p* = 0.26	*r* = −0.81; *p* = 0.10	*r* = 0.67; *p* = 0.33
BS, kg	*r* = −0.32; *p* = 0.54	*r* = 0.20; *p* = 0.71	*r* = 0.54; *p* = 0.27	*r* = 0.52; *p* = 0.30	*r* = −0.84; *p* = 0.17
BS, rbw	*r* = −0.20; *p* = 0.71	*r* = 0.12; *p* = 0.83	*r* = 0.62; *p* = 0.19	*r* = 0.35; *p* = 0.50	*r* = −0.89; *p* = 0.11
FS, kg	*r* = −0.66; *p* = 0.16	*r* = 0.59; *p* = 0.22	*r* = 0.82; *p* = 0.05 *	*r* = 0.81; *p* = 0.05 *	*r* = −0.95; *p* = 0.05 *
FS, rbw	*r* = −0.57; *p* = 0.24	*r* = 0.52; *p* = 0.29	*r* = 0.90; *p* = 0.02 *	*r* = 0.69; *p* = 0.13	*r* = −0.92; *p* = 0.08
Snatch, kg	*r* = −0.90; *p* = 0.02 *	*r* = 0.90; *p* = 0.01 *	*r* = 0.75; *p* = 0.09	*r* = 0.91; *p* = 0.01 *	*r* = −0.88; *p* = 0.12
Snatch, rbw	*r* = −0.90; *p* = 0.02 *	*r* = 0.92; *p* = 0.01 *	*r* = 0.85; *p* = 0.03 *	*r* = 0.89; *p* = 0.02 *	*r* = −0.81; *p* = 0.19
C&J, kg	*r* = −0.89; *p* = 0.02 *	*r* = 0.86; *p* = 0.03 *	*r* = 0.72; *p* = 0.11	*r* = 0.93; *p* = 0.01 *	*r* = −0.94; *p* = 0.06
C&J, rbw	*r* = −0.90; *p* = 0.01 *	*r* = 0.89; *p* = 0.02 *	*r* = 0.89; *p* = 0.02 *	*r* = 0.92; *p* = 0.01 *	*r* = −0.87; *p* = 0.13
TS, kg	*r* = −0.78; *p* = 0.07	*r* = 0.72; *p* = 0.11	*r* = 0.80; *p* = 0.06	*r* = 0.90; *p* = 0.01 *	*r* = −0.96; *p* = 0.04 *
TS, rbw	*r* = −0.71; *p* = 0.11	*r* = 0.68; *p* = 0.14	*r* = 0.93; *p* = 0.01 *	*r* = 0.80; *p* = 0.06	*r* = −0.90; *p* = 0.10
IT, N.m	*r* = 0.13; *p* = 0.81	*r* = 0.09; *p* = 0.87	*r* = 0.13; *p* = 0.81	*r* = −0.39; *p* = 0.44	*r* = 0.66; *p* = 0.34
IT, rbw	*r* = 0.12; *p* = 0.82	*r* = 0.09; *p* = 0.87	*r* = 0.30; *p* = 0.56	*r* = −0.37; *p* = 0.57	*r* = 0.48; *p* = 0.52
Tibana test, rep	*r* = −0.96; *p* < 0.01 *	*r* = 0.98; *p* < 0.01 *	*r* = 0.71; *p* = 0.11	*r* = 0.84; *p* = 0.04 *	*r* = −0.63; *p* = 0.38

VO_2max_, maximal oxygen consumption; BS, back squat; FS, front squat; C&J, clean and jerk; TS, total strength; IT, isometric torque; rbw, relative to body weight; * *p* < 0.05.

## Data Availability

Data sharing is not applicable.

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
