# Peer review of "Local Muscle Endurance and Strength Had Strong Relationship with CrossFit® Open 2020 in Amateur Athletes"

_sports, 2021, doi:10.3390/sports9070098_

Round 1
Reviewer 1 Report
L29. Change “A local…” to “Local…”.
L75. Please have correct units for relative maximum oxygen uptake (throughout the paper) and mention in L75 the parameter. Therefore, cange “ml/kg/min- 1” to “mL·(kg·min)- 1”. Ensure in Table 3 and 4 to have a middle dot as multiplication sign.
L75. Please provide the unit for relative back squat muscle strength.
L88. I suggest to change “the following study” to “the present study”.
Figure 1. Change “VO2” to “VO2max”.
Figure 1. Change “suqat” to “squat”.
L103. Change “an in vivo variation coefficient” to “coefficient of variation”.
L113. Please clarify what is meant by “both tests”.
L134. Please clarify what is meant by “both testing sessions”.
L165. Please define “WODs”.
L168 and throughout the paper convert values in lb to values in kg.
Table 2. Isometric strength units should be I think N/m as it is torque. I suggest to change “Isometric strength” to “Isometric torque”. Also make unit change in Tables 3 and 4.
L251. Change “the Open.” to “the CrossFit® Open 2020.” In addition, ensure consistency and use “CrossFit® Open 2020” throughout the paper.
L291. Body fat values are taken to categorize not being overweight. However, body mass index is used to define weight categories. Please revise as height was seemingly not measured and body fat mean and SD value cannot be used to make a statement about the weight categorization of the participants.
L365. Change “14. 14. “ to “14.“
Author Response
Reviewer 1
We appreciate the time and effort that the reviewer dedicated to providing feedback on our manuscript and are grateful for the insightful comments and valuable improvements to our paper. We have performed a deep change in the manuscript according to your advice (outlined in the yellow marked version).
L29. Change “A local…” to “Local…”.
Response: We have changed it as required.
L75. Please have correct units for relative maximum oxygen uptake (throughout the paper) and mention in L75 the parameter. Therefore, cange “ml/kg/min- 1” to “mL·(kg·min)- 1”. Ensure in Table 3 and 4 to have a middle dot as multiplication sign.
Response: We have changed it as required.
L75. Please provide the unit for relative back squat muscle strength.
Response: We have added it as required.
L88. I suggest to change “the following study” to “the present study”.
Response: We have changed it as required.
Figure 1. Change “VO2” to “VO2max”.
Response: We have changed it as required.
Figure 1. Change “suqat” to “squat”.
Response: We have changed it as required.
L103. Change “an in vivo variation coefficient” to “coefficient of variation”.
Response: We have changed it as required.
L113. Please clarify what is meant by “both tests”.
Response: The word "both" was incorrect. We substitute by word "all"
L134. Please clarify what is meant by “both testing sessions”.
Response: The word "both" was incorrect. We substitute by word "all"
L165. Please define “WODs”.
Response : We define the word “WOD” in the line 38
L168 and throughout the paper convert values in lb to values in kg.
Response : Thank you for your important comment. We have changed it as required.
Table 2. Isometric strength units should be I think N/m as it is torque. I suggest to change “Isometric strength” to “Isometric torque”. Also make unit change in Tables 3 and 4.
Response : Thank you for your important comment. We have changed it as required.
L251. Change “the Open.” to “the CrossFit® Open 2020.” In addition, ensure consistency and use “CrossFit® Open 2020” throughout the paper.
Response: Thank you for your important comment. We have changed it as required. We ensure consistency and use “CrossFit® Open 2020” throughout the paper.
L291. Body fat values are taken to categorize not being overweight. However, body mass index is used to define weight categories. Please revise as height was seemingly not measured and body fat mean and SD value cannot be used to make a statement about the weight categorization of the participants.
Response: We agree with your suggestion, and we reformulated the text. First, we added in the “Methods Section” how the body weight was measured (line 108). Second, we reformulated our sentence in the Discussion.
Discussion:
Line 329: An interesting finding of the current study is that athletes displayed a low body fat per-centage (men 11.9 ± 6.0; woman 8.8 ± 5.0). Possibly, athletic performance is more affected by excessive body fat percentage, as demonstrated in previous studies [18,19,20].
L365. Change “14. 14. “ to “14.“
Response : We have changed it as required.
Reviewer 2 Report
The primary objective of the study was analyze the relationship between anthropometric measures, cardiorespiratory capacity, and the variables of strength, power, and a specific test of muscular endurance with performance in the CrossFit® Open 2020.
Introduction:I suggest adding some information about crossfit, training methods, exercises and training and starting loads, the course of the competition and how to prepare for the start. In this way, the aim of the research will be emphasized even more.
The article is generally well written following the steps of the scientific method, based on solid literature, methods are clear, detailed and reproducible, and results are well presented and discussed in relation to the literature. The methodical part is written clearly and correctly
When using abbreviations, make sure you explain them first, then use the correct abbreviation throughout the article.
I suggest to explain the abbreviations used, eg RM, AMRAP and WOD.
The number of respondents is a bit small in order to draw firm generalizations, it is worth pointing out in the limitations of the study.
In my opinion, tables 1 and 2 should be corrected. The results of women and men should not be added up and mixed up.
Line 240: VO2max subscript
I propose to divide the conclusions into theoretical ones, i.e. those resulting directly from the research, and the application ones, based on assumptions and own thoughts.
In general, my concerns about the reliability of the entire research are raised by a small number of respondents. I am interested in how the authors will explain this fact.
Therefore, conclusions may refer only to this particular group, they should not be generalized.
Author Response
Reviewer 2
The primary objective of the study was analyze the relationship between anthropometric measures, cardiorespiratory capacity, and the variables of strength, power, and a specific test of muscular endurance with performance in the CrossFit® Open 2020.
Response: We appreciate the time and effort that the reviewer dedicated to providing feedback on our manuscript and are grateful for the insightful comments and valuable improvements to our paper. We have performed a deep change in the manuscript according to your advice (outlined in the yellow marked version).
Introduction: I suggest adding some information about crossfit, training methods, exercises and training and starting loads, the course of the competition and how to prepare for the start. In this way, the aim of the research will be emphasized even more.
Response: We agree with your comments and we added more details about crossfit, training methods, exercises and training and starting loads. The course of the competition and how to prepare for the start was reported in the second paragraph of the introduction.
Line 35 : CrossFit® is considered as a constantly varied, high-intensity, functional movement training aiming at increasing work capacity across multiples physical domains (endurance, strength, flexibility) using several functional movements [1]. Therefore, different types of workout sessions, which are known as Workout of the Day (WOD), combines different exercises and tasks, such as cycling, running, rowing, olympic weightlifting, power weightlifting and gymnastic-type exercises [1].
The article is generally well written following the steps of the scientific method, based on solid literature, methods are clear, detailed and reproducible, and results are well presented and discussed in relation to the literature. The methodical part is written clearly and correctly.
Response : Thank you for your important comment
When using abbreviations, make sure you explain them first, then use the correct abbreviation throughout the article.
Response : Thank you for your important comment. We have changed it as required.
I suggest to explain the abbreviations used, eg RM, AMRAP and WOD.
Response : Thank you for your important comment. We have changed it as required.
The number of respondents is a bit small in order to draw firm generalizations, it is worth pointing out in the limitations of the study.
Response: We agree with your comments and we highlight in the limitations (line 331)
Line 331: First, the findings are limited to a relatively small (n = 17) sample of convenience, our specific athlete characteristics and time frame.
In my opinion, tables 1 and 2 should be corrected. The results of women and men should not be added up and mixed up.
Response : Thank you for your important comment. We have changed it as required.
Line 240: VO2max subscript
Response : Thank you for your important comment. We have changed it as required.
I propose to divide the conclusions into theoretical ones, i.e. those resulting directly from the research, and the application ones, based on assumptions and own thoughts.
Response : Thank you for your important comment. We have changed it as required.
Line 342: In conclusion, we report for the first time that a specific muscle endurance test and strength had strong relationship with CrossFit® open workout performance. In contrast to our hypothesis, we did not observe any correlation between anthropometric profile and cardiorespiratory capacity with CrossFit® Open performance.
From a practical perspective, coaches and practitioners can use these findings to assess physical fitness and organize the distribution of the training session based on less developed physical needs, in order to ensure an appropriate physiological adaptation for a given competition. In future studies, the focus should be on researching on larger samples, different training levels, as well the holistic assessment model of athletes.
Response: Thank you for your important comment The limitations were rephrased as requested. We make it clear to the reader that the conclusions may refer only to this particular group, they should not be generalized.
Line 330 : Despite the interesting findings of this study, some limitations need to be mentioned. First, the findings are limited to a relatively small (n = 17) sample of convenience, our specific athlete characteristics and time frame. Second, the specific test (Tibana) does not include all gymnastics (e.g., ring and bar muscle-ups, rope climbs, pistols etc.), weightlifting (e.g., snatch) and powerlifting exercises (e.g., deadlift) that are usually performed at the CrossFit® Open. Third, it should be noted, that these results should be considered only for amateur athletes from a specific region and competition. Therefore, our findings may not be directly transferable to elite athletes or others experience, which relativizes the generalization and extrapolation of the obtained findings. Finally, the cross-sectional nature of this study prevents the ability to identify any causal relationship between study variables and outcomes.
Therefore, conclusions may refer only to this particular group, they should not be generalized.
Response: Thank you for your important comment The limitations were rephrased as requested. We make it clear to the reader that the conclusions may refer only to this particular group, they should not be generalized.
Line 330 : Despite the interesting findings of this study, some limitations need to be mentioned. First, the findings are limited to a relatively small (n = 17) sample of convenience, our specific athlete characteristics and time frame. Second, the specific test (Tibana) does not include all gymnastics (e.g., ring and bar muscle-ups, rope climbs, pistols etc.), weightlifting (e.g., snatch) and powerlifting exercises (e.g., deadlift) that are usually performed at the CrossFit® Open. Third, it should be noted, that these results should be considered only for amateur athletes from a specific region and competition. Therefore, our findings may not be directly transferable to elite athletes or others experience, which relativizes the generalization and extrapolation of the obtained findings. Finally, the cross-sectional nature of this study prevents the ability to identify any causal relationship between study variables and outcomes.
Round 2
Reviewer 1 Report
L22. Change “and power…” to “power…”
L24. Change “Specific test…” to “Specific tests…”
L61. Change “test” to “tests”
L150. Change “25. ºC” to “25 °C”
L162. Delete “index”
Table 1. Change “26.63” to “26.6”
L243. Change “…tables 3” to “…table 3”
Table 4. Change “IT, kg” to “IT, N/m”
L277. Change “test” to “tests”
Author Response
We appreciate the time and effort that the editor and reviewers dedicated to providing feedback on our manuscript and are grateful for the insightful comments and valuable improvements to our paper. We have performed all changes in the manuscript according to reviewers advice (outlined in the yellow marked version).
Reviewer 2 Report
Accept in present form
Author Response

(The authors gave the same response as above.)
